# Determining the Core Competencies of a Coach: Design and Validation of a New Methodology

**DOI:** 10.3390/bs13010018

**Published:** 2022-12-24

**Authors:** Zuzana Birknerova, David Misko, Lucia Zbihlejova

**Affiliations:** 1Department of Managerial Psychology, Faculty of Management and Business, University of Presov, 080 01 Presov, Slovakia; 2Department of Intercultural Communication, Faculty of Management and Business, University of Presov, 080 01 Presov, Slovakia

**Keywords:** coaching, core competencies, methodology, factor analysis, questionnaire

## Abstract

Coaching is a complex and diverse profession with a wide range of uses. Despite determining the approach to coaching, examining the core competencies of coaches is rarely explored. The aim of this study was to design and validate the CCC3 methodology (the three core competencies of a coach). The research sample consisted of 387 respondents for the exploratory factor analysis and 532 respondents for the confirmatory factor analysis. The confirmatory factor analysis proved the existence of three factors that cover a coach’s core competencies, namely, F1—effective communication and awareness creation, F2—relationship creation and active listening, and F3—laying the foundations. The study also provides an overview of the applicability of the CCC3 methodology in coaching practice. Exploratory and confirmatory factor analysis made it possible to create a 24-item model that describes the core competencies of a coach. The verified questionnaire can be used as a tool to assess the key, i.e., the core competencies of a coach, which represent the basis for the coaching approach, enriching the issue of coaching on a practical level. It is possible to use the methodology in management and education but also wherever it is necessary to evaluate the core competencies of a coach.

## 1. Introduction

Coaching is one of the methods of development of skills and abilities of a person. It is a targeted strengthening of the development potential of an individual, enabling the maximization of their performance. It works according to the principle of increasing the internal commitment and self-improvement of this individual [1]. The essence of coaching is thinking, which needs to be fundamentally changed. Individuals working as managers (or in related areas) can use coaches to encourage or help them to change their behavior or leadership style, and only then they can achieve the desired changes in their team [2].

The International Coaching Federation [3] defines coaching as “partnering with clients in a thought-provoking and creative process that inspires them to maximize their personal and professional potential”. According to Whitmore [4], the pioneer of coaching, coaching unblocks a person’s potential and maximizes their performance. It does not teach people but helps them understand how to learn. Coaching can be understood as a relationship that helps in setting goals, plans, and solutions. It directs the development and performance of the coachee. The interest in coaching in the work sector has recently increased as it can be applied in the business development and introduction of new products and services. Coaching is a key part of supporting personnel to increase their effectiveness and the use of their potential [5].

The consensus of definitions creates a theoretical structure of coaching, which is focused at least on theses such as the use of potential, increasing performance, personal and career growth, creating goals and their implementation, breaking down internal barriers, motivation, and more. To understand coaching, it is also necessary to master the theoretical and practical comparison of coaching and related areas such as mentoring, psychotherapy, and counselling [3,6,7,8,9,10,11,12], and other contexts in which coaching is highly applicable, for instance, the managerial context (e.g., [13,14,15]). Coaches’ key or core competencies are an important predictor of their skills and knowledge. The main objective of the paper is, therefore, to verify the methodology for the assessment of the core competencies of aa coach. Factor extraction and the confirmatory factor analysis allowed us to compile a questionnaire and verify its suitability. A validated questionnaire for assessing aa coach’s core competencies represents one of the tools for the development of coaching. A similar methodology designed to assess coaching as an effective managerial tool was previously developed by Birknerová et al. [16] and further perfected by Birknerová, Zbihlejová, and Frankovský [17]. It was labelled as AC (Assessment of Coaching) and consists of 18 items grouped (after a factor analysis) into four factors, namely, success, relationships, decision-making, and performance.

Coaching as a profession is covered by the International Coaching Federation (ICF), which defines and describes the key competencies that form the basis of a coach’s training and are assessed in the coach’s accreditation. These competencies are based on practical experience from the coaching processes [3]. The importance of these competencies is also emphasized by Alhashmi [18], who regards them as a basic prerequisite for coach accreditation and mastering the coaching approach. Gavin [19] describes the key competencies, their importance, and the need for their development in accordance with ICF. Despite the general discussion that there are other competencies of a coach, a coach’s key competencies are often treated as an indisputable part of using a coaching approach and the education of coaches as defined by ICF ([20,21,22,23,24]).

### 1.1. Core Competencies of a Coach

According to the ICF model [3], the core competencies are grouped into four clusters. The priority of individual competencies is not evaluated as they are all key to demonstrating the knowledge and skills of a coach. The Global Council of ICF [3] has recently approved a revised competency model, which entered into force in January 2021. According to this model, the four clusters and eight core competencies of a coach are as follows:


Foundation


Demonstrates ethical practice;Embodies a coaching mindset.


Co-Creating the Relationship


3.Establishes and maintains agreements;4.Cultivates trust and safety;5.Maintains presence.


Communicating Effectively


6.Listens actively;7.Evokes awareness.


Cultivating Learning and Growth


8.Facilitates client growth.

AA coach’s first competency is related to understanding and adhering to a code of ethics. Apart from the theoretical definition, a coach must not use an approach during coaching that is aimed at counselling, therapy, or the promotion of one’s own opinion. In line with the second competency, extensive studies in psychotherapy have shown that the most important predictor of a client’s outcome is a coach–client relationship. With a positive approach, it provides optimal conditions for growth, where clients accept and take responsibility for themselves. Any auxiliary relationship presupposes change [25]. A coach’s presence is not only understood as a physical presence. Siegel [26] describes the presence, not necessarily a physical one, but the very presence or present, which is not affected by a coach’s past, future, or personal needs. The third competency represents active listening as a higher level of listening, where full attention is focused on the client, avoiding distractions and actually paying attention to the speaker’s language [27,28,29,30]. Wallis [31] describes his first experience with coaching as a relatively simple process based on listening and actively asking questions. De Haan [32] states that the aim of coaching should be to deepen the situation through reflective questions. The fourth competency manifests a revelation that coaching creates the conditions for learning and behavior change, which characterizes it as a continuous cycle of education [33]. Similarly, coaching is described as a continuous cycle of activity and learning that together create change [34].

### 1.2. Research Questions

Based on the abovementioned theory, two research questions were formulated and further investigated:Is it possible to specify the internal factor structure of the proposed model of the CCC3 methodology?Is it possible to create a model that would confirm the existence of the tested structure?

The following parts of the paper present the research, the main focus of which is to provide answers to these issues.

## 2. Materials and Methods

As already previewed, the aim of the paper is to verify the questionnaire for the assessment of the core competencies of a coach. The core competencies are an important predictor of a coach’s skills and knowledge. The proposed CCC3 (the three core competencies of a coach) methodology was inspired by the theory and categorization of the core coaching competencies according to the ICF [3] as presented above. It was created as a self-assessment tool. The respondents were asked to assess the degree to which they reflect (i.e., have) the individual competencies. The methodology is thus suitable not only for coaches, managers, or trainers but also for the respondents with a tendency to use coaching as a tool, regardless of the industry or work position. CCC3 can be also used in education, e.g., to conduct ante and post measurements, evaluate the effectiveness of education, or track changes in the core coaching competencies in the course of education.

The CCC3 questionnaire consists of 24 items related to the core competencies of a coach. The original draft contained 33 items formulated on the basis of the core coach competences’ structure [3], but after performing the factor analysis and detecting the Cronbach’s alpha reliability indicators, only 24 items were proved to be statistically reliable and valid. Verification was performed on the Slovak version of the questionnaire (both linguistic versions were composed by the authors of the methodology). Items were assessed on a scale from 1 to 6 (1 = definitely not; 6 = definitely yes). 

To validate the questionnaire, responses from 387 respondents, of which 208 (53.75%) were women and 179 (46.25%) were men, were collected in early 2021. The average age of the respondents was 33.6 years. Despite the absence of the use of random selection, we tried to reach the proportionality of the research sample from different perspectives, such as gender, age, and residence region. Data collection was carried out using the questionnaire method via Google Forms. The research was conducted on a sample of Slovak respondents. Data were collected from the authors’ freely available contact database. According to job classification, the research sample consisted of 90 accredited coaches, 77 managers, 100 entrepreneurs, and 120 employees, from a wide range of industries.

The analysis was performed using the following software: IBM SPSS 26, IBM SPSS AMOS 26, and STATISTICA.

## 3. Results

By means of a factor analysis (using the principal component method with varimax rotation), three factors, which indicate and describe the individual core competencies, were extracted and labeled as F1—effective communication and awareness creation, F2—relationship creation and active listening, and F3—laying the foundations (Figure 1, Table 1a,b).

Cattell’s scree plot shows a number of factors found, but only the first three factors are sufficiently saturated. For these factors, the Eigenvalues were greater than 1.

The extracted factors explain 68.649% of the variance. The suitability of using factor analysis is confirmed by the Kaiser–Mayer–Olkin measure of sampling adequacy (0.970), the Barlett sphericity test (*p* = 0.000), and the KMO assessment rate, calculated as cross-correlations of variables using anti-image matrices. We determined the reliability of the questionnaire by assessing the Cronbach’s Alpha values:F1—Effective communication and awareness creation: α = 0.946; number of items = 10,F2—Relationship creation and active listening: α = 0.906; number of items = 8,F3—Laying the foundations: α = 0.943; number of items = 6.

The differences in the assessment of the three areas of a coach’s competencies are also confirmed by Friedman’s two-way analysis of variance (*p* = 0.000; decision: reject the null hypothesis that the three factors are identical). A closer assessment of the individual factors and a comparison of the research studies provided us with additional information that supports the fact that the competencies we propose can be defined in terms of their content (Table 2).

After assessing the Skewness and Kurtosis measures, we concluded that the data did not have the perfect shape of a normal distribution (the skewness and kurtosis values indicate the normality of the data distribution). If these two measures range from −1 to 1, the distribution of values in the dataset should be regarded as symmetric [35]. It is also possible to take into account the statement of the authors [22,36], who claim that at a research sample size above 200, the data are considered to be normally distributed. The proposed structure of the defined factors of the CCC3 methodology is also supported by the values of the calculated intercorrelation coefficients between the individual factors (Table 3), which we obtained using the Pearson correlation coefficient.

Table 3 indicates that individuals scoring high in one of the three core competency factors will have a higher level of the other two as well. Respondents who can:Ask effective questions that encourage others to think,Lead others to seek new ideas,Help others discover broader contexts,Create opportunities for the growth of others,Encourage others to discover their potential,Help others make plans,Create a plan that is measurable and feasible,Obtain all of the necessary information to create a comprehensive plan,Transfer rights and responsibilities to others,and develop the ability of others to make their own decisions.They are also able to:Provide support to others,Show genuine interest in others,Use an open and confidential style of communication,Be always present in the conversation and pay attention to the context,Listen and respect others,Pay attention to the non-verbal elements of communication,and listen to someone actively and intently without interrupting.

Simultaneously, these individuals understand the principles of coaching, know how to apply the principles of coaching, know the coach’s code of ethics, can assess whether there can be a match between their methods and the needs of others, are able to understand what is expected in a coaching relationship, and can specify the responsibilities of the coach and the client.

### Confirmatory Factor Analysis

We tested the presented factor structure by means of a confirmatory factor analysis (CFA). Three models were used for the test: the unidimensional model with one factor, the correlated model with three factors, and the higher-order factor model with four factors, using the *maximum likelihood* estimation (MLE). Confirmatory factor analysis was performed on a sample of 532 respondents, of which 282 were women and 250 were men. The average age of the respondents was 35.3 years. In terms of employment, the sample consisted of 100 accredited coaches, 121 managers, 150 entrepreneurs, and 161 employees from various industries.

Model suitability indicators such as Chi-square, the root mean square error of approximation (RMSEA), the goodness of fit index (GFI), and the comparative fit index (CFI) were used to assess the suitability of the tested models (more about these indicators in, e.g., [37,38,39]).

The assessment of these indicators suggests that the unidimensional and the higher-order models significantly exceeded the model’s suitability thresholds. The confirmatory factor analysis confirmed the suitability of the model with three interrelated factors. The tested model showed the possibility of improving the suitability indicators by adding covariance between some residues (Figure 2, Table 4).

Based on these indicators, the model proves suitable. Χ^2^ was significant in all of the models, which is a common phenomenon in larger samples. The Χ^2^/df ratio did not exceed an acceptable limit of two to five. The given ratio has the value of Χ^2^/df = 3.63. The GFI, CFI, and RMSEA indicate a close match. The regression weights were significant and positively associated in our model. This indicates that individuals who reach a higher value in one of the three factors of the core competencies of a coach have a higher value in the other two as well.

Thus, the two research questions formulated in the beginning of the research may be answered as follows:It is possible to specify the internal factor structure of the proposed model of the CCC3 methodology.It is possible to create a model that would confirm the existence of the tested structure.

## 4. Discussion

The ICF identifies eight key coaching competencies, which are based on the experience and research of practitioners and academicians. The purpose is always to ensure that the ICF competencies reflect the evolving practice and growing understanding of the behavioral and psychological processes involved in coaching. The analysis carried out in 2018/2019 [40] was based on a thorough two-year analysis of the coaching practice. In this study, the research sample consisted of 1300 coaches from around the world, representing a diverse range of coaching styles. It is this in-depth analysis of the revised model of key competencies that is considered to be the “gold standard” in coaching qualifications. The revised model contains virtually all existing concepts of the previous model and therefore confirms the validity of these competencies [3,40]. Our model provides us with a holistic view of the core competencies of a coach. Developing all three factors of a coach’s core competencies should be an integral part of the personal and professional development of any coach in order to achieve the best possible results. When accrediting coaches within the framework of the International Coaching Federation, these core competencies are a requirement to obtain the title of an accredited coach (more in [3]). However, the core competencies are not only important in terms of accreditation and professional coaching, they are an integral part of anyone who wants to use the coaching approach in everyday life. This approach is applicable in management and in education, as well as in everyday work or family relationships. Although the person-centered approach was presented by Rogers as early as 1940 [41], coaching is very specific. The coaching profession can be considered to be the fastest growing professions in the 21st century. This indicates a real need for coaching, i.e., a coaching approach in various areas. People are not computers, so they need conditions and space to utilize what they really know. Due to an increasing need for high-quality staff, applicability of students and solution to common problems, more than an autocratic approach to peopleis required.

## 5. Conclusions

The CCC3 methodology can be used within various settings and at least in the following areas: the self-assessment of an individual in the core competencies of a coach for the needs of starting a job or the further development of the competencies of a coach; the evaluation of employees, coaches, and students by another person, e.g., a supervisor, a professional coach, or a teacher; ex ante and ex post measurements, e.g., in determining the effectiveness of coaching courses; and the possibility of further use for scientific research purposes, e.g., for other statistical analyses, such as a correlation analysis using a different methodology or variables such as gender, age, or personality traits.

As for the limitations of this study, the item translation and the subjectivity of the researcher could have caused errors in the estimated model. As the data collection by means of the CCC3 methodology was carried out on the sample of Slovak respondents using the Slovak version of the questionnaire, the CFA model was created by means of the information collected. It would be, therefore, appropriate to carry out another CFA and create a model based on the data collection conducted using the English version of the methodology. Similarly, the results could be skewed by the sample size, but all of these deficiencies will be corrected in the next stage of our research.

## Figures and Tables

**Figure 1 behavsci-13-00018-f001:**
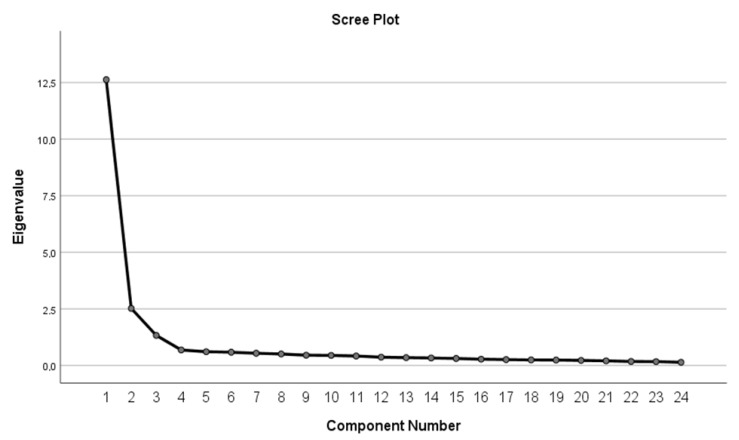
Scree plot of the extracted factors.

**Figure 2 behavsci-13-00018-f002:**
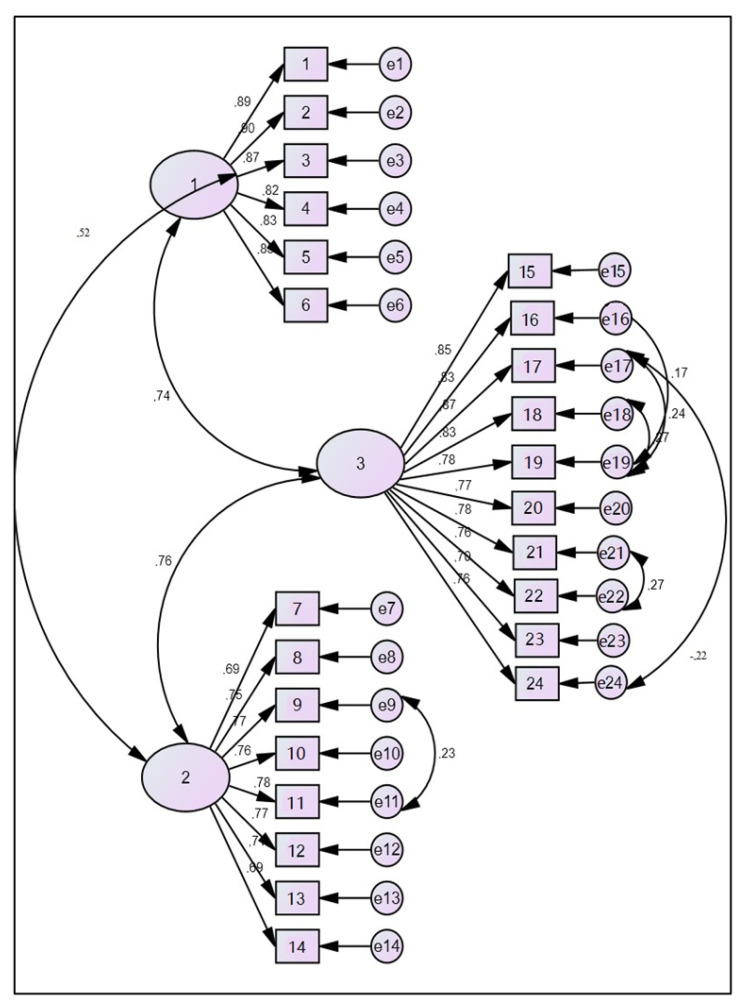
The three-correlated-factors model with 24 items (1 = F3; 2 = F2; and 3 = F1).

**Table 1 behavsci-13-00018-t001:** (a). Factor structure of the CCC3 methodology. (b) Eigenvalues and variance explained.

**(a)**							
**Questionnaire Items**			**F1** **Effective Communication and Awareness Creation**		**F2** **Relationship Creation** **and Active Listening**		**F3** **Laying the Foundations**
1. I understand the principles of coaching.							0.842
2. I know how to apply the principles of coaching.							0.843
3. I know the coach’s code of ethics.							0.850
4. I can assess whether there can be a match between my methods and the needs of others.							0.743
5. I am able to understand what is expected in a coaching relationship.							0.778
6. I can specify the responsibilities of a coach and a client.							0.770
7. I show genuine interest in others.					0.625		
8. I provide support to others.					0.754		
9. In communication, I can fully focus on others.					0.776		
10. I use an open and confidential style of communication.					0.728		
11. I am always present in the conversation and pay attention to the context.					0.737		
12. I am able to listen and respect others.					0.794		
13. I pay attention to the non-verbal elements of communication (tone of voice, body language).					0.618		
14. I can listen to someone actively and intently without interrupting.					0.683		
15. I can ask effective questions that encourage others to think.			0.643				
16. I can lead others to seek new ideas.			0.722				
17. I help others discover broader contexts.			0.742				
18. I create opportunities for the growth of others.			0.706				
19. I encourage others to discover their potential.			0.694				
20. I help others make plans.			0.687				
21. I am able to create a plan that is measurable and feasible.			0.690				
22. I can obtain all the necessary information to create a comprehensive plan.			0.762				
23. I can transfer rights and responsibilities to others.			0.656				
24. I develop the ability of others to make their own decisions.			0.647				
**(b)**							
**Factor**	**Eigenvalue**	**Extraction Sums of Squared Loadings**			**Rotation Sums of Squared Loadings**		
		**Total**	**% of Variance**	**Cumulative %**	**Total**	**% of Variance**	**Cumulative %**
F1	12.624	12.624	52.600	52.600	5.961	24.836	24.836
F2	2.521	2.521	10.504	63.105	5.378	22.409	47.245
F3	1.331	1.331	5.545	68.649	5.137	21.404	68.649

**Table 2 behavsci-13-00018-t002:** Descriptive statistics.

Factor	N	Min	Max	Mean	SD	Skewness	Std. Error	Kurtosis	Std. Error
F1—Effective communication and awareness creation	387	1	6	4.220	1.082	−0.345	0.124	−0.263	0.247
F2—Relationship creation and active listening	387	1	6	4.692	0.953	−0.824	0.124	0.823	0.247
F3—Laying the foundations	387	1	6	3.877	1.340	−0.259	0.124	−0.740	0.247

**Table 3 behavsci-13-00018-t003:** Intercorrelations of the CCC3 methodology factors.

Factors		F1—Effective Communication and Awareness Creation	F2—Relationship Creation and Active Listening
F2—Relationship creation and active listening	*r*	0.708	-
	*p*	0.000	-
F3—Laying the foundations	*r*	0.706	0.501
	*p*	0.000	0.000

**Table 4 behavsci-13-00018-t004:** Indicators of suitability of the tested model.

	Χ^2^	df	*p*	GFI	CFI	RMSEA
**Tested model**	574.703	243	0.000	0.959	0.994	0.05
**Limit values**			≥0.05	≥0.90	≥0.95	≤0.06

## Data Availability

Due to privacy and ethical restrictions, data is available upon request only.

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
