# Peer review of "Determining the Core Competencies of a Coach: Design and Validation of a New Methodology"

_behavsci, 2022, doi:10.3390/bs13010018_

Round 1
Reviewer 1 Report
In this research, based on a sample of several hundred respondents of both sexes from Slovakia, the authors tried to design, verify and validate the methodology of assessing the basic competencies that a successful coach should have. They proposed the CCC3 (Three Core Competencies of the Coach) model, based on a completed 24-item questionnaire where each question contains a six-point scale, which was inspired by the theory and categorization of the ICF model of core competencies. The questionnaire was analyzed by factor analysis, whereby three factors stood out and the authors labelled them as; 1) Effective communication and awareness creation, 2) Relationship creation and active listening, and 3) Laying the foundations.
The authors should explain in more detail what guided them in choosing the 24 questions they included in the questionnaire - why exactly 24 questions, why not more or less?
At first I thought that the authors created and verified the CCC3 questionnaire originally in the Slovak language and later translated it into English, to include it in this paper and make it available to others. But in the conclusions, authors reported that “.. the item translation ... could have caused errors in estimated model”.
General comments
Authors often quote someone's work without mentioning the first authors author's name, such as " According to [4], the pioneer of coaching, …“ (line 36) or „[19] describes the key competences …“ (line 65). Please avoid such citations - either mention the last name of the first author of the study you are citing, or form the sentence so that it contains the core finding of the study and at the end mention the reference number.
When describing the respondents, in addition to trainers, managers and entrepreneurs, the authors stated that among them were "120 employees from various branches of industry". It is not clear to me from which position they answered the questions in the questionnaire? Were they instructed to answer how important they consider certain characteristics of a coach/motivator or did they answer for themselves, e.g. do they „lead others to seek new ideas“ (question 16), even though they are not coaches themselves?
Title – please change it to be more informative, I suggest a title „Determining the key competencies of a coach: design and validation of a new methodology“
Abstract
line 10 – the authors mentioned that „coaching is a rich profession“ – not clear if they meant rich as in making money or rewarding/satisfying?
lines 10-12, please rephrase: „Despite determining the approach to coaching, examining the core competencies of coaches is rarely explored.“
line 19 – „The verified questionnaire …“
line 20 – „ … a tool to assess the key competences …“
line 21 – „… coaching on a practical…“
Keywords
please change the word „factors“ to „factor analysis“, word „coach“ to „coaching“ and add a new keyword: „questionnaire“
Introduction
lines 31-33 – the way this sentence is written makes it sound like managers can change their behavior, not that coaches should encourage clients/managers (or help them) to change their behavior or leadership style to achieve the desired changes in their team
lines 42-43 – please change to „… part of supporting personnel to increase their effectiveness and use …“
line 71 – please add „According to the ICF model, the core competencies are grouped into four clusters.“
In lines 74-75, the authors point out that there are eight competencies. In line 88, the sentence begins by emphasizing that it is about and describing the first competence. As the sentences that follow do not state that they describe the second, third and other competencies, it seems that they also describe the first competency. To make it clear to the readers, please indicate in each sentence which competence the description refers to
line 96 – please change to „… but the very presence or present, which is …“
lines 97-98 – please change to „… where full attention is focused on the client, avoiding distractions and actually paying attention to the speaker's language.“
line 101 – „reflective questions“
line 102 – „behavior change“
Results
line 142 – please change „sig.“ to „p=“
line 149 – the same comment as the previous one
lines 150-155 – these two sentences do not describe the results of your research, delete them from this section
lines 173-182 - this sentence is too long, please rewrite it into more sentences
line 197 – please change „X2“ to „Chi-square“
Discussion
lines 221-227 – I assume that these four sentences refer to the same research [41], so please provide that reference immediately after the first sentence and then link that first sentence to the following sentences using demonstratives (e.g. that analysis, that revised model)
lines 228-229 – it is not clear what the authors wanted to say with this sentence: “Developing all three factors of the coach's core competencies should be an integral part of any coach …“
Lines 230-231 - please change to “Coaches, these core competencies are necessary for ...”
Conclusion
The authors stated that the “... the item translation ... could have caused errors in estimated model”. I do not understand this comment, isn`t your questionnaire written in Slovak and filled out by Slovaks?
Author Response
Dear Reviewer,
thank you very much for the thorough review. We have discussed and amended the manuscript accordingly (please, see are attached response).
Authors

Reviewer 2 Report
It is suggested to supplement the article with research questions or hypotheses at the beginning and to answer them in the final part of the article. This could significantly increase the scientific dimension of the article, which is very important due to the issues raised.
Author Response
Dear Reviewer,
thank you very much for the feedback. We have added the research questions to the paper.
Authors